# Genetic insights into *Cyphocharax magdalenae* (Characiformes: Curimatidae): Microsatellite *loci* development and population analysis in the Cauca River, Colombia

Ana Maria Ochoa-Aristizábal[☯], Edna Judith Márquez[ID]*[☯]

Laboratorio de Biología Molecular y Celular, Escuela de Biociencias, Facultad de Ciencias Universidad Nacional de Colombia – Sede Medellín, Medellín, Antioquia, Colombia

☯ These authors contributed equally to this work.
* ejmarque@unal.edu.co

## Abstract

*Cyphocharax magdalenae*, a Colombian freshwater fish species, plays a vital role in nutrients distribution and serves as a significant food source for other fish species and local fishing communities. Considered a short-distance migratory species, *C. magdalenae* populations face substantial extinction risk due to human activities impacting their habitats. To address the lack of knowledge on genetic diversity and population structure, this study used next-generation sequencing technology to develop species-specific microsatellite *loci* and conducted a population genetics analysis of *C. magdalenae* in the middle and lower sections of the Cauca River, Colombia. Out of 30 pairs of microsatellite primers evaluated in 324 individuals, 14 *loci* were found to be polymorphic, at linkage equilibrium and, in at least one population, their genotypic frequencies were in Hardy-Weinberg equilibrium. Results showed high genetic diversity levels compared to other neotropical Characiformes, with inbreeding coefficients similar to those reported for phylogenetically related species. Moreover, *C. magdalenae* exhibits seasonal population structure (rainy-dry) consisting of two genetic stocks showing bottleneck signals and high effective population sizes. This information is essential for understanding the current species genetics and developing future management programs for this fishery resource.

## Introduction

The field of ichthyology is marked by a comprehensive fish diversity examination, with 36,775 valid species cataloged to date, 18,688 of which thrive exclusively in freshwater environments [1]. Within this diverse assemblage, the order Characiformes, a significant facet of Ostariophysi, stands out with a notable count of 2,334 valid species confined to freshwater habitats [2]. This taxonomically rich group spans the continents of Africa, South America, Central America, and southern North America, concentrating approximately 90% of its diversity in the Neotropical region [2]. Within Characiformes, the Curimatidae family encompasses 9

**Data Availability Statement:** All relevant data are within the manuscript and its Supporting information files.

**Funding:** This study was supported by a grant framed under the Project "Variabilidad genética de un banco de peces de los sectores medio y bajo del río Cauca" (CT-2019-000661, Empresas Públicas de Medellín and Universidad Nacional de Colombia, Sede Medellín). Funders do not play any role in the study design, data collection and analysis, decision to publish, or preparation of the manuscript.

**Competing interests:** The authors have declared that no competing interests exist.

genera and 121 valid species distributed in the Neotropics [1]. *Cyphocharax* Fowler 1906 genus is one of the most abundant, having 47 valid species [1] and wide distribution ranging from the Pacific rivers in southern Costa Rica to the La Plata River and various coastal drainages in central Argentina [3, 4]. It has been found that Atlantic basins have a much more diverse group expanding from the Orinoco River, Amazonas River, rivers from the Guayanas, and eastern Brazil; on the contrary, a less diverse group of species habits rivers from northern Peru to Costa Rica, comprising the tributaries of Magdalena and Maracaibo [5].

*Cyphocharax magdalenae* (Steindachner 1878) commonly known as "viejito" or "Yalúa" (Fig 1) distributes from the Coto River in southeastern Cosa Rica [6] and Panamá [7, 8], to the Maracaibo lake and its tributaries in Venezuela [9] and Colombia [10]. In Colombia, *C. magdalenae* is the only species of the genus reported for the Atrato, Sinú, Canalete and Magdalena-Cauca basin [10, 11]. The Cauca River, main tributary of the Magdalena-Cauca basin [12, 13], has great economic importance due to its influence on most of the prioritized productive chains in Colombia, which include small exploitations, extensive industries, monocultures, energy generation, mineral extraction, agriculture and fishing [12, 14, 15]. This has resulted in fish being one of the most threatened biological resources, with few or unknown local priorities on rivers [16].

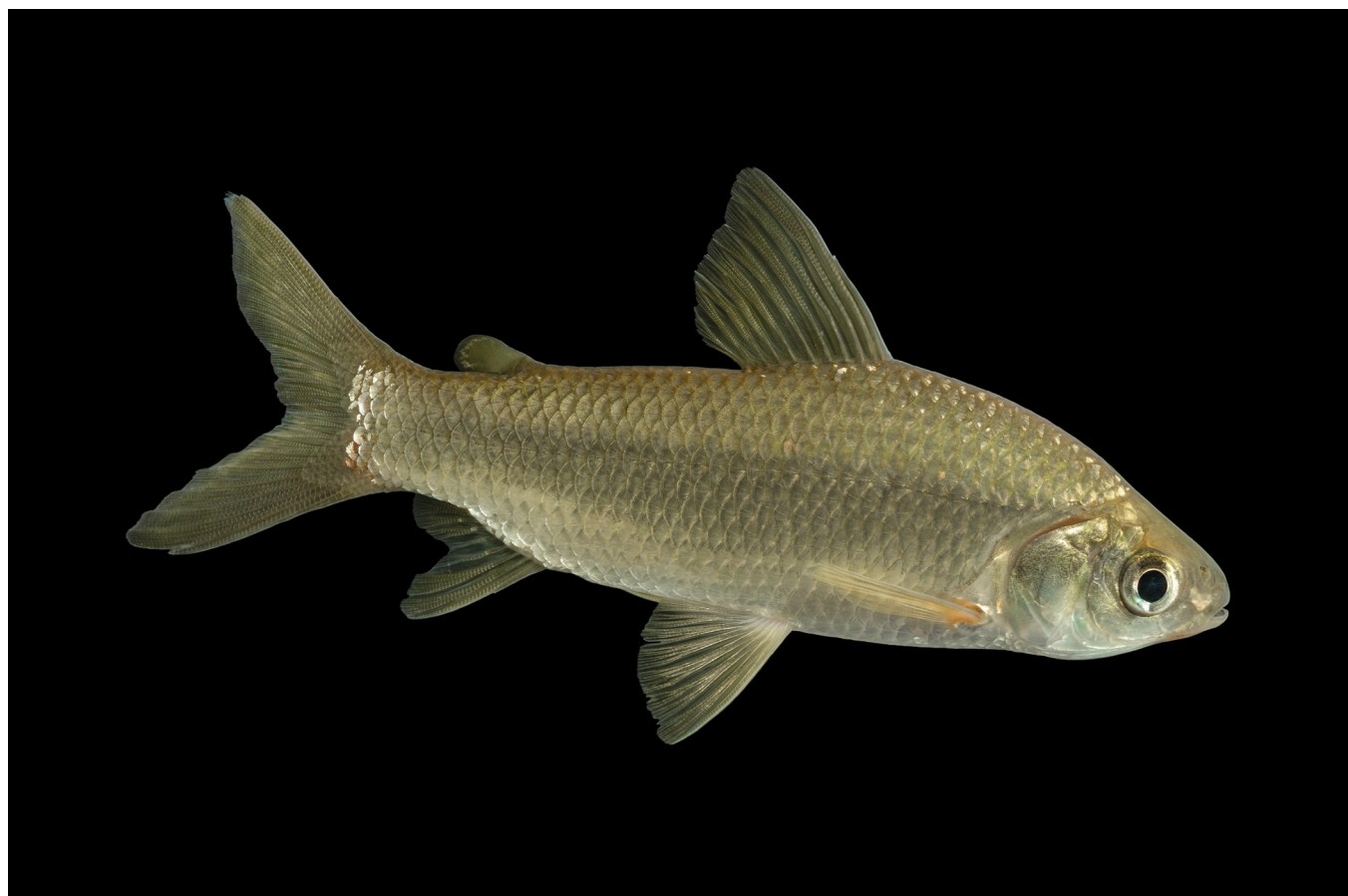

**Fig 1. Photography of *Cyphocharax magdalenae*, a neotropical freshwater fish.** Courtesy of Jorge E. García-Melo and Luis J. García-Melo, Proyecto CaVFish Colombia.

*Cyphocharax magdalenae* constitutes 30% of the total population within the lower Magdalena-Cauca floodplains and plays a crucial role in energy, nutrients, coal, and minerals transportation and sediment resuspension in the Cauca basin [15]. This potamodromous species undergoes short-distance reproductive migrations (<100 km) [17], spawns twice a year [11, 15], and lays small (0.73 ± 0.04 mm) adhesive eggs with small perivitelline space [11, 18]. Moreover, this fish has a generation time of 2.6 years [19], lacks parental care for embryos, and does not build nests.

Additionally, the fishing importance of *C. magdalenae* has been increasing during the last years due to the replacement of traditionally captured species such as catfish *Prochilodus magdalenae* Steindachner 1879 and Blanquillo *Sorubim cuspicaudus* Littmann, Burr & Nass 2000 [15, 20]. Although classified as a Least Concern species by The International Union for Conservation of Nature (IUCN), its notable overfishing-caused mortality suggests a possible decrease in its populations [11, 21, 22]. Moreover, *C. magdalenae* lacks genetic studies that allow knowing the current species status and delving into its ecological processes, which, in turn, hinders the species protection and conservation plans development and application.

The scarce information for *C. magdalenae* contrasts with population genetic studies performed for *Curimata mivartii* [23] and species with fishing relevance of the phylogenetically close family Prochilodontidae like *Prochilodus magdalenae* [24–26], *Prochilodus reticulatus* [27] and *Ichthyoelephas longirostris* [28]. Likewise, it contrasts with studies on species of other families inhabiting the Magdalena-Cauca basin such as *Pseudoplatystoma magdaleniatum* [29, 30], *Brycon henni* [31–33], *S. cuspicaudus*, *Ageneiosus pardalis*, *Pimelodus grosskopfii* [34], *Pimelodus yuma* [35], *Pseudopimelodus magnus* and *Pseudopimelodus atricaudus* [36]. Among the different markers used in the cited studies, microsatellites stand out as the most used tool to solve questions at micro-evolutive scale in population genetics. In this area, the Single Nucleotide Polymorphisms have been utilized for species delimitation and for exploring genetic introgression in *Pimelodus* species [37].

To estimate the species genetic status in the lower section of the Cauca River, this study contrasted three hypotheses: (i) *C. magdalenae* shows signs of genetic deterioration with low genetic variability levels, high inbreeding levels and recent bottlenecks resulting from habitat alterations due to different anthropogenic activities; (ii) the species exhibit high gene flow as a result of the scarce or non-existent geographic barriers in this habitat, which is based on evidence found for other Curimatidae species [23]; and (iii) seasons influencing fish migration in the Cauca River shape the population structure of *C. magdalenae*. To test these hypotheses, this study identified de novo and characterized species-specific microsatellite *loci* to *C. magdalenae*. This approach was chosen instead of using heterologous *loci* to avoid potential pitfalls associated with cross-amplification [38], such as allele size homoplasy [39], unsuccessful amplification in phylogenetically distant species [40], low polymorphism, presence of null alleles [41], and amplification of non-orthologous *loci* [42].

## Materials and methods

### Biological material and study area

This study analyzed a total of 324 muscle tissues and/or fins of *C. magdalenae* preserved in ethanol 70%, provided between 2019 and 2021 by Universidad de Antioquia, Universidad de Córdoba, and Universidad Nacional de Colombia Sede Medellín, through scientific cooperation agreement CT-2019-000661, under environmental license # 0155 of January 30th, 2009, from Ministry of Environment, Housing and Territorial Development for the Ituango hydroelectric construction.

Samples from individuals were collected in 19 sites of the main channel, some swamps, and tributaries of the lower sections of the Cauca River. This zone, characterized by a large alluvial plain surrounded by mountains and flat and wavy surfaces where various swampy complexes are formed [43, 44], has been exposed to different anthropic activities including stockbreeding, fishing and extensive cultures [see 23]. Additionally, high mercury and sediment levels and elevated vegetation amounts are reported in the main courses of the river as a result of deforestation and flooding that may affect ichthyofauna [45]. According to their distribution, samples were grouped in five sections (S4-S8, Fig 2) previously described by Landínez-García & Márquez (2016) [28].

## Microsatellite *loci* identification and development of primers

Methodology previously described [23, 28] was followed for developing species-specific microsatellite *loci* for *C. magdalenae*. The genomic library was built from total DNA extraction from the fin (5,775 µg) of a *C. magdalenae* individual captured in the Cauca River (CYPHO11886) using the DNA extraction reagents and recommendations of the manufacturer PureLink genomic DNA Mini Kit (Invitrogen), then it was used for NGS surface sequencing (Whole Genome Sequencing), through Illumina Miseq (300 PE). After raw reads cleaning using PRINSEQ lite (removal of adapter sequences utilizing <Q30 quality bases), 100,000 high-quality extended reads were analyzed with PAL_FINDER v0.02.03 [46] for extracting reads containing tri-, tetra- and pentanucleotide microsatellite motifs. Then, Primer3 v2.0 [47] was used for the primer designs in the flanking sequences of the microsatellite *loci* and ultimately the correct evaluation of the primers was tested with ePCR [48]. Selection of the microsatellite *loci* set was performed according to already stablished features for validating new primers of microsatellite *loci* [49, 50].

For genotyping, 10 µl reaction mixtures were used with final concentrations of 0.3 pmol/µl of each forward primer tagged on the 5' end with one of four adapters (tails A, B, C and D; [51]), 6 pmol/µl of each reverse primer, 0.5 pmol/µl fluorescently labeled adapter (6-FAM, VIC, NED and PET, Applied Biosystems), 1X Master Mix, 2.5% v/v GC Enhancer Platinum Multiplex PCR Master Mix (Applied Biosystems) and 3–5 µg/µl DNA isolated using the Purelink® purification kit (Thermo Scientific) with a modification in digestion time of 24 h when working with fins. Thermal profiles included an initial denaturalization step at 90˚C for 35 s and an annealing step at 56˚C for 35 s (with no final elongation). Subsequently, amplicons were separated by electrophoresis on an ABI 3730 XL automated sequencer using 600 LIZ as the internal molecular size. Then, GeneMarker® v3.0.0 was employed to denote allele fragments according to their molecular size (100 to 450 pb) and Micro-Checker v.2.2.3 [52] to detect potential genotyping errors.

## Polymorphism, genetic diversity, outlier *loci* detection, and demographical events

Polymorphism information content (PIC) and average number of alleles per *locus* for each microsatellite marker were calculated using Cervus v3.0.7 [53] and GenAlEx v6.5.03 [54]. Estimation of observed ($H_O$) and expected ($H_E$) heterozygosities, inbreeding coefficient ($F_{IS}$) and tests for departures from Hardy-Weinberg equilibrium were performed using Arlequin v3.5.2.2 [55], applying the sequential Bonferroni correction for the statistical significance in the multiple comparisons [56, 57].

For detecting outlier *loci* and determining evolutive forces acting over the microsatellite *loci*, BayeScan v2.1 [58] was used for performing an analysis employing the parameters for prior odds of 10:1 for the neutral model, 20 pilot runs, each having 5,000 iterations, followed

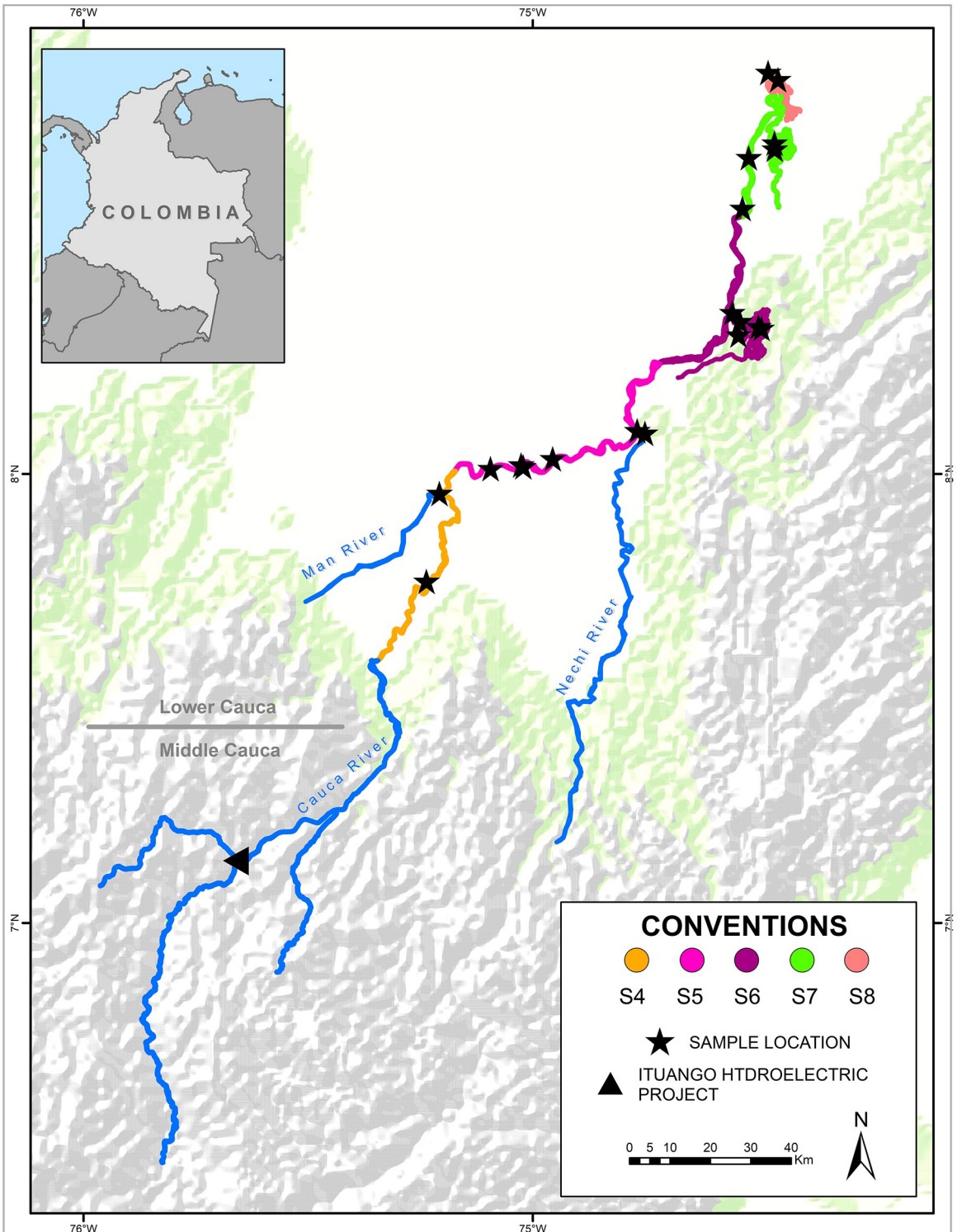

**Fig 2. Sampling sites (stars) of *Cyphocharax magdalenae* in the lower section of the Cauca River (S4-S8).** Self-made creation of the map based on contour lines scaled 1:100,000 from the Instituto Geográfico Agustín Codazzi source, 2019 (Available from: IGAC Geoportal, https://geoportal.igac.gov.co/contenido/datos-abiertos-cartografia-y-geografia).

by 100,000 iterations and a burn-in of 50,000. Critical values were based on the posterior probability of the Bayes factor, following the Jeffrey's scale [59], which sets a probability of 0.76 as substantial evidence for selection.

To determine recent genetic bottlenecks, excess heterozygosity was tested under the mutation drift equilibrium assumption in three mutational models of the microsatellite *loci* (IAM: infinite alleles model, SMM: stepwise mutational model, TPM: two-phase model) through Wilcoxon signed rank test [60] included in Bottleneck v.1.2.02 [61]. Moreover, M ratio (mean ratio of the number of alleles to allele size range); [62] included in Arlequin v3.5.2.2 [55] was calculated following the criteria according to which values lower than 0.68 indicate recent and severe reductions in population size [62]. Furthermore, effective population size ($N_e$) of the species was determined in each of the assessed populations using the linkage disequilibrium method and a minimum allele frequency of 0.02 implemented in NeEstimator v2.1 [63]. Change in $H_E$ of the species in 10, 50 and 100 generations (t) was estimated based on the equation $Ht/H0 = [1 - 1/(2Ne)]^t$ [64]. Genetic deterioration grade was stablished using the critical values of 25% Ht reduction in 10 (critically in danger), 50 (in danger) and 100 (vulnerable) generations [65].

### Genetic structure

Bayesian analysis of population partitioning in STRUCTURE v.2.3.4 [66] was performed to determine samples grouping according to their co-ancestry coefficient. Parameters included 1,000,000 steps of the Markov chain Monte Carlo; 100,000 iterations were regarded as burn-in to estimate each K value (1–8). Each analysis was repeated 20 times. For better estimating the genetic stocks (K), STRUCTURESELECTOR [67] was used for calculating the ΔK ad hoc statistic [68], estimators MEDMEANK, MAXMEANK, MEDMEDK and MAXMEDK [69], and to graphically represent the results using the integrated CLUMPAK software [70].

Genetic differentiation of *C. magdalenae* among sections, genetic groups suggested by STRUCTURE, and seasons (rainy and dry) was calculated using the standardized statistics $F_{ST}$[71–73] and Jost's $D_{EST}$ [74, 75], and Analysis of Molecular Variance (AMOVA) [73] using GenAlEx v6.503 software [54]. Additionally, a Discriminant Analysis of Principal Components (DAPC) was performed in polymorphic *loci* genotype resulting in 324 analyzed individuals using the R- package Adegenet [76]. Moreover, allele frequency distributions in genetic stocks were assessed through a G-test in Genepop v. 4.7.5. software [77].

## Results

### Identification of microsatellite *loci* and detection of outlier *loci*

Low-coverage genome sequencing raw data provided a total of 3,224,415,544 base pairs (bp), 10,712,344 reads, 41.28%GC and 83.72% Q30. After depuration, 100,000 reads (42,621,284 bp) were obtained, 37,119 out of which had compound repeat motifs, 1,054 interrupted repeat motifs and 71,376 contained microsatellite *loci* with direct repetitions (53,649 potentially amplifiable *loci*). Furthermore, 30,120 were validated using electronic PCR.

Out of 30 tested *loci*, 16 showed clearly defined peaks and absence of stutter bands in the electropherograms; these microsatellite *loci* include tri- (3) and tetra- (13) nucleotides motifs. Micro-Checker analysis did not indicate stutter-associated genotyping errors nor allele loss in the *loci* (dropout), and linkage disequilibrium tests were not significant. Departures of allele frequencies from Hardy-Weinberg equilibrium and heterozygote deficits were identified (Table 1). In general, microsatellite *loci* exhibited allele size between 106 and 414 bp with PIC values ranging from 0.733 to 0.954 and $H_O$ and $H_E$ between 0.408–0.916 and 0.762–0.957,

**Table 1. Primer sequences and features of 16 microsatellite *loci* selected for *Cyphocharax magdalenae*.**

| Locus | Motif | Primer sequence (5'– 3') | Range (bp) | Na | PIC | $H_O$ | $H_E$ | $P_{HWE}$ |
|---|---|---|---|---|---|---|---|---|
| Cym03 | (AAAG)n | F:TGCACTTAAATCCTGTCCATAAGC R:TCTGGATTGAGGCTTGAACG | 212–264 | 13 | 0.733 | 0.716 | 0.762 | 0.000** |
| Cym04 | (ATCT)n | F:TGAGATTCATAAATTGGGAAGAGG R:AAAGGCATGTCTGTGTTGCC | 230–298 | 18 | 0.898 | 0.840 | 0.908 | 0.010* |
| Cym06 | (AAAT)n | F:TCATCTTCCATTATTTGTGCTGG R:GAGGAACGATTTACCCATGC | 224–324 | 23 | 0.920 | 0.819 | 0.926 | 0.000** |
| Cym07 | (ATT)n | F:AGGCTCATTAGTACCAGCAGC R:TGTGCACAACAGGAACATCC | 111–195 | 27 | 0.941 | 0.861 | 0.945 | 0.000** |
| Cym08 | (ATAC)n | F:ACCAAACTGGAAGACAGCCC R:TAAACAACTTCCACACCGGC | 216–332 | 29 | 0.939 | 0.765 | 0.943 | 0.000** |
| Cym10 | (TTC)n | F:TTACTGCGAGCGGACCTACC R:CAGCCAACATGGTTTGTATGG | 236–407 | 40 | 0.929 | 0.647 | 0.935 | 0.000** |
| Cym11 | (AAAT)n | F:AACCCAAGCCTTCAATCAGG R:TGACCAACCGTATCCCTGC | 280–312 | 9 | 0.771 | 0.691 | 0.797 | 0.000** |
| Cym14 | (ATCT)n | F:TGTATGGTTTTGTCATTTGCACC R:CCTCAGGGAGTTATTTGCCG | 215–375 | 35 | 0.954 | 0.842 | 0.957 | 0.000** |
| Cym16 | (ATCT)n | F:GAAAGGGCAAATAACCTACTGC R:ATACCGGTTGTGCTGTGACC | 254–414 | 34 | 0.928 | 0.408 | 0.934 | 0.000** |
| Cym18 | (ATGG)n | F:CAGGGCAAATACTGCCTTCC R:TCCACAAAGAAGGCCACTCC | 185–289 | 27 | 0.922 | 0.873 | 0.928 | 0.023* |
| Cym19 | (TCTG)n | F:TGGCGTATAAACATCAGCGG R:CTGGAACTGCCAAAGATCCC | 157–225 | 17 | 0.846 | 0.851 | 0.861 | 0.530 |
| Cym24 | (ATT)n | F:GCTGGACTTCCATATCCACTCG R:GCTATTCTTCCCCTGCATCG | 250–325 | 26 | 0.946 | 0.910 | 0.950 | 0.011* |
| Cym25 | (AAAT)n | F:CCCTGCCTTCATTAGCATCC R:CCAGCCTTCCTCTTTCTCCC | 106–198 | 23 | 0.910 | 0.843 | 0.918 | 0.000** |
| Cym26 | (AAAG)n | F:CAACTCGGAGTGACCTACCG R:CTGTGTGGACCGAGTGTTCC | 125–197 | 19 | 0.900 | 0.833 | 0.909 | 0.000* |
| Cym29 | (AATG)n | F:ATCTGGCAGTTTGTCCAGGG R:CAGTGTCCCAGACCGAGACC | 190–234 | 12 | 0.791 | 0.792 | 0.816 | 0.114 |
| Cym30 | (AGTG)n | F:GACGTTCCATCGCTTCATCC R:GAGAGAGTCTGACTGATGACCAGC | 184–236 | 14 | 0.751 | 0.763 | 0.782 | 0.600 |

Na number of alleles per locus; PIC polymorphic information content; $H_O$ Observed heterozygosity; $H_E$ Expected heterozygosity; $P_{HWE}$ P value of tests for departure from Hardy-Weinberg equilibrium (α: 0.05);

*P < 0.05;

**P < 0.01.

respectively (Table 1); no outlier *loci* were detected by Bayescan. Nonetheless, null alleles were found in *loci* Cym10 and Cym16; thus, they were excluded in subsequent analyses.

## Genetic diversity and demographic events

Genetic diversity estimators (Table 2) showed that S7 had the lower average number of alleles per *locus* (15.714 alleles/*locus*) while S5 exhibited the higher mean number (18.857 alleles/*locus*). Mean $H_O$ was lower in S7 (0.772) and higher in S8 (0.827) and mean $H_E$ displayed the lower value in S5 (0.880) and the higher in S4 (0.887). Only two (Cym8 and Cym14) out of the 14 selected *loci* departed from the Hardy-Weinberg equilibrium in four of the five evaluated sections and the remaining fulfilled the assumption in at least two sections. A similar result of genetic diversity was observed in the genetic stocks (Stock1 and Stock2, Table 2) found by the genetic structure Bayesian analysis, as shown below.

**Table 2. Genetic diversity of *Cyphocharax magdalenae* in five sections (S4, S5, S6, S7 and S8) of the Cauca River in Colombia.**

| Pop | | Cym19 | Cym 4 | Cym3 | Cym29 | Cym25 | Cym7 | Cym24 | Cym14 | Cym8 | Cym30 | Cym11 | Cym6 | Cym18 | Cym26 | Across loci |
|---|---|---|---|---|---|---|---|---|---|---|---|---|---|---|---|---|
| **S4** | Na | 14 | 15 | 11 | 9 | 19 | 22 | 24 | 27 | 21 | 11 | 8 | 19 | 19 | 14 | 16.643 |
| | $H_O$ | 0.956 | 0.822 | 0.773 | 0.778 | 0.711 | 0.800 | 0.886 | 0.841 | 0.791 | 0.844 | 0.667 | 0.795 | 0.800 | 0.889 | 0.811 |
| | $H_E$ | 0.878 | 0.910 | 0.756 | 0.801 | 0.920 | 0.947 | 0.950 | 0.957 | 0.924 | 0.781 | 0.830 | 0.903 | 0.926 | 0.912 | 0.885 |
| | $P_{HWE}$ | 0.584 | 0.269 | 0.117 | 0.019* | 0.000** | 0.004** | 0.069 | 0.002** | 0.147 | 0.854 | 0.010* | 0.131 | 0.080 | 0.195 | 0.000** |
| | $F_{IS}$ | -0.090 | 0.098 | -0.023 | 0.029 | 0.229 | 0.157 | 0.068 | 0.123 | 0.145 | -0.083 | 0.199 | 0.120 | 0.137 | 0.026 | 0.076* |
| **S5** | Na | 14 | 15 | 12 | 9 | 22 | 26 | 24 | 32 | 27 | 12 | 9 | 20 | 24 | 18 | 18.857 |
| | $H_O$ | 0.796 | 0.860 | 0.652 | 0.811 | 0.936 | 0.849 | 0.926 | 0.883 | 0.766 | 0.772 | 0.702 | 0.796 | 0.883 | 0.777 | 0.815 |
| | $H_E$ | 0.844 | 0.913 | 0.695 | 0.803 | 0.929 | 0.943 | 0.947 | 0.958 | 0.951 | 0.792 | 0.819 | 0.917 | 0.907 | 0.900 | 0.880 |
| | $P_{HWE}$ | 0.613 | 0.467 | 0.273 | 0.696 | 0.237 | 0.083 | 0.626 | 0.000** | 0.000** | 0.444 | 0.000** | 0.001** | 0.066 | 0.004** | 0.000** |
| | $F_{IS}$ | 0.057 | 0.058 | 0.062 | -0.010 | -0.008 | 0.099 | 0.023 | 0.078 | 0.195 | 0.025 | 0.143 | 0.133 | 0.027 | 0.138 | 0.067* |
| **S6** | Na | 14 | 15 | 10 | 9 | 18 | 25 | 25 | 31 | 25 | 12 | 9 | 20 | 24 | 16 | 18.071 |
| | $H_O$ | 0.883 | 0.853 | 0.750 | 0.753 | 0.844 | 0.896 | 0.948 | 0.792 | 0.787 | 0.727 | 0.727 | 0.805 | 0.883 | 0.882 | 0.824 |
| | $H_E$ | 0.875 | 0.903 | 0.788 | 0.814 | 0.913 | 0.939 | 0.953 | 0.957 | 0.941 | 0.785 | 0.772 | 0.926 | 0.941 | 0.909 | 0.887 |
| | $P_{HWE}$ | 0.698 | 0.019* | 0.306 | 0.116 | 0.006** | 0.091 | 0.039* | 0.000** | 0.004** | 0.208 | 0.000** | 0.004** | 0.172 | 0.131 | 0.000** |
| | $F_{IS}$ | -0.009 | 0.055 | 0.048 | 0.075 | 0.076 | 0.046 | 0.005 | 0.173 | 0.165 | 0.074 | 0.058 | 0.131 | 0.061 | 0.030 | 0.066* |
| **S7** | Na | 12 | 15 | 11 | 10 | 15 | 23 | 21 | 24 | 20 | 10 | 9 | 17 | 18 | 15 | 15.714 |
| | $H_O$ | 0.865 | 0.806 | 0.811 | 0.806 | 0.730 | 0.811 | 0.778 | 0.838 | 0.622 | 0.722 | 0.568 | 0.806 | 0.865 | 0.784 | 0.772 |
| | $H_E$ | 0.848 | 0.917 | 0.794 | 0.815 | 0.894 | 0.947 | 0.944 | 0.950 | 0.933 | 0.779 | 0.790 | 0.933 | 0.915 | 0.923 | 0.884 |
| | $P_{HWE}$ | 0.129 | 0.064 | 0.743 | 0.412 | 0.062 | 0.001** | 0.001** | 0.163 | 0.000** | 0.297 | 0.001** | 0.022* | 0.053 | 0.061 | 0.000** |
| | $F_{IS}$ | -0.020 | 0.123 | -0.021 | 0.012 | 0.186 | 0.146 | 0.179 | 0.120 | 0.337 | 0.074 | 0.284 | 0.138 | 0.056 | 0.152 | 0.120* |
| **S8** | Na | 16 | 18 | 12 | 11 | 21 | 23 | 23 | 29 | 23 | 12 | 7 | 20 | 25 | 16 | 18.286 |
| | $H_O$ | 0.817 | 0.829 | 0.676 | 0.814 | 0.859 | 0.901 | 0.930 | 0.845 | 0.800 | 0.761 | 0.718 | 0.887 | 0.901 | 0.845 | 0.827 |
| | $H_E$ | 0.866 | 0.898 | 0.788 | 0.820 | 0.904 | 0.940 | 0.951 | 0.953 | 0.950 | 0.773 | 0.722 | 0.929 | 0.936 | 0.903 | 0.881 |
| | $P_{HWE}$ | 0.416 | 0.102 | 0.001** | 0.854 | 0.019* | 0.041* | 0.786 | 0.000** | 0.000** | 0.737 | 0.719 | 0.436 | 0.519 | 0.007** | 0.000** |
| | $F_{IS}$ | 0.057 | 0.078 | 0.143 | 0.007 | 0.050 | 0.041 | 0.023 | 0.114 | 0.159 | 0.017 | 0.006 | 0.045 | 0.037 | 0.064 | 0.058* |
| **Stock1** | Na | 15 | 15 | 12 | 9 | 22 | 25 | 25 | 33 | 26 | 13 | 9 | 19 | 25 | 16 | 18.857 |
| | $H_O$ | 0.895 | 0.856 | 0.696 | 0.768 | 0.860 | 0.850 | 0.956 | 0.886 | 0.805 | 0.723 | 0.702 | 0.830 | 0.886 | 0.816 | 0.823 |
| | $H_E$ | 0.861 | 0.899 | 0.740 | 0.783 | 0.893 | 0.939 | 0.942 | 0.956 | 0.945 | 0.774 | 0.785 | 0.906 | 0.926 | 0.896 | 0.875 |
| | $P_{HWE}$ | 0.796 | 0.764 | 0.046* | 0.524 | 0.099 | 0.353 | 0.960 | 0.000** | 0.002** | 0.204 | 0.000** | 0.157 | 0.031* | 0.013* | 0.000** |
| | $F_{IS}$ | -0.039 | 0.048 | 0.060 | 0.020 | 0.038 | 0.096 | -0.015 | 0.073 | 0.148 | 0.065 | 0.106 | 0.084 | 0.043 | 0.090 | 0.051* |
| **Stock2** | Na | 17 | 18 | 12 | 12 | 23 | 26 | 24 | 32 | 29 | 14 | 7 | 23 | 26 | 19 | 20.143 |
| | $H_O$ | 0.828 | 0.832 | 0.726 | 0.806 | 0.833 | 0.867 | 0.885 | 0.818 | 0.743 | 0.785 | 0.686 | 0.813 | 0.867 | 0.842 | 0.809 |
| | $H_E$ | 0.860 | 0.910 | 0.773 | 0.815 | 0.896 | 0.942 | 0.948 | 0.956 | 0.941 | 0.786 | 0.696 | 0.930 | 0.928 | 0.888 | 0.876 |
| | $P_{HWE}$ | 0.905 | 0.028* | 0.026* | 0.280 | 0.078 | 0.070 | 0.023* | 0.000** | 0.000** | 0.839 | 0.771 | 0.001** | 0.003** | 0.061 | 0.000** |
| | $F_{IS}$ | 0.038 | 0.086 | 0.061 | 0.011 | 0.070 | 0.080 | 0.067 | 0.145 | 0.211 | 0.002 | 0.015 | 0.125 | 0.066 | 0.051 | 0.071* |

Na number of alleles per locus; $H_O$ y $H_E$ Observed and expected heterozygosities, respectively; $F_{IS}$ inbreeding coefficient; $P_{HWE}$ P statistical significance for the Hardy-Weinberg test (α: 0.05).

*P < 0.05;

**P < 0.01.

Moreover, the inbreeding coefficients across *loci* in each population showed statistical significance with greater impact in population S7 ($F_{IS}$ = 0.120, P = 0.000). These values, although remained significant, significantly decreased in the stocks ($F_{IS}$Stock1 = 0.051, P = 0.000; $F_{IS}$Stock2 = 0.071, P = 0.000).

It was found that assessed populations recently suffered a drastic reduction in population size since both the modified Garza-Williamson index (M-ratio: 0.238–0.256) and P values of

**Table 3. Bottleneck tests and effective population size (N$_e$) of *Cyphocharax magdalenae* in the five sampling sections and genetic stocks of the Magdalena-Cauca basin.** Statistical significance is noted in bold (P<0.05).

| Pop | IAM[1] | SMM[1] | TPM[1] | M-ratio[2] | N$_e$[3] |
|---|---|---|---|---|---|
| S4 | 0.000** | 0.879 | 0.052 | 0.238 | 316 (166.4–1997.1) |
| S5 | 0.000** | 0.852 | 0.021* | 0.256 | ∞ (1194.3–∞) |
| S6 | 0.000** | 0.620 | 0.000** | 0.248 | 1869.2 (470.5–∞) |
| S7 | 0.000** | 0.932 | 0.015* | 0.243 | 223.9 (116.1–1571.6) |
| S8 | 0.000** | 0.966 | 0.039* | 0.250 | 233590.8 (692.8–∞) |
| Stock1 | 0.000** | 0.988 | 0.052 | 0.252 | 1464.6 (567.7–∞) |
| Stock2 | 0.000** | 0.979 | 0.002** | 0.257 | ∞ (2127–∞) |
| Overall | 0.000** | 0.891 | 0.000** | 0.262 | (1314.5–10252.9) |

[1] P values of the Wilcoxon signed rank test of one tail in IAM, SMM and TPM;

[2] M-ratio < 0.68 denote recent reductions in the population;

[3] IC 95% using the Jackknife method (Waples & Do, 2008).

* P < 0.05;

**P < 0.01

the Wilcoxon signed rank test for the IAM and the TPM showed statistical significance (Table 3). Results for SMM, however, were not significant.

Furthermore, the effective size calculation in each sampled section showed the lower values in S4 and S7 and the higher in S6 and S8; nevertheless, it was not possible to obtain a value for S5, and estimation was obtained only for one of the genetic stocks (Stock1) (Table 3). Due to the latter, in subsequent analyses the lower limits of confidence intervals were used for S5 and Stock2.

For all evaluated generations, the species is classified as non-threatened as no reduction percentage reached 25% (critical value of the classification) (Table 4). S8 showed the lower reduction percentage in all generations (0.002, 0.011, 0.021); nevertheless, S7 and S4 exhibited the greater loss of heterozygosity percentage in 100 generations with 20.034 and 14.645, respectively. Moreover, in the evaluation on genetic stocks, no approximation surpassed the 5% reduction in all generations; in fact, similar values were found in each stock for each generation, although Stock1 showed slightly higher values.

## Genetic structure

It was found that the most likely number of populations (ΔK) based on the Bayesian inference approach was 2. These two groups coexist in all sampled sections as confirmed by the co-ancestry histogram (Fig 3A). Additionally, it was observed that Stock1 predominates in sections S4 and S5 (pink), while Stock2 (purple) predominates in sections S6, S7 and S8. This was corroborated by the DAPC, showing that the two genetic stocks were not homogeneously distributed (Fig 3B and 3C); only just one similarity was found in the distribution of the tested individuals in sections S6 and S7 (Fig 3B). Differences in the frequencies of the stocks agree with results obtained in AMOVA (F′$_{ST (4, 647)}$ = 0.003, P = 0.001) and the pairwise comparisons of the standardized statistics F'$_{ST}$, D$_{EST}$ (Table 5), which showed significant differences between S8 in relation to the remaining sections, and S6 in relation to S4 and S5.

Genetic structure comparison by collect season (rainy: April-June, October-December; dry: January-March, July-August; Serna et al., 2021) and sampling years, in addition to confirming the coexistence of two genetic stocks within the evaluated sections of the Cauca River (ΔK = 2,

**Table 4. Estimation of the change in $H_E$ of *Cyphocharax magdalenae* in 10, 50 and 100 generations.**

| Pop | t | $H_t$ reduction | $H_t$ reduction (%) |
|---|---|---|---|
| S4 | 10 | 0.014 | 1.571 |
| | 50 | 0.067 | 7.612 |
| | 100 | 0.130 | 14.645 |
| S5 | 10 | 0.004 | 0.418 |
| | 50 | 0.018 | 2.072 |
| | 100 | 0.036 | 4.101 |
| S6 | 10 | 0.002 | 0.267 |
| | 50 | 0.012 | 1.329 |
| | 100 | 0.023 | 2.640 |
| S7 | 10 | 0.020 | 2.211 |
| | 50 | 0.093 | 10.576 |
| | 100 | 0.177 | 20.034 |
| S8 | 10 | 0.000 | 0.002 |
| | 50 | 0.000 | 0.011 |
| | 100 | 0.000 | 0.021 |
| Stock1 | 10 | 0.003 | 0.341 |
| | 50 | 0.015 | 1.693 |
| | 100 | 0.029 | 3.357 |
| Stock2 | 10 | 0.002 | 0.235 |
| | 50 | 0.010 | 1.169 |
| | 100 | 0.020 | 2.324 |

Mean LnP(K) = -22763.31000, MEDMEANK, MAXMEANK, MEDMEDK = 2; Fig 4A),
revealed that during rainy seasons, the capture percentage was greater for Stock2 during the
sampled years (2020: 91.7%; 2021: 94.3%), while Stock1 was primarily captured during dry sea-
son (2020: 67%; 2021: 58.8%) in comparison to the moderated captures of Stock2 (2020: 33%;
2021: 41.2%) (Fig 4B).

## Discussion

To test three hypotheses related to the population genetics of *C. magdalenae*, this study devel-
oped a set of 16 polymorphic microsatellite *loci* capable of detecting diversity levels and genetic
structure in the lower section of the Cauca River. The selected microsatellite *loci* were prefera-
bly those having tri- and tetranucleotide repetition motifs as these repetition patterns are the
most recommended for their simplicity for genotyping and allele classification [46, 78, 79]. All
microsatellite *loci* exhibited PIC values that allow describing them as highly informative
according to Bostein et al., 1980 [80]; moreover, these values were higher than those reported
for *C. mivartii* (0.549–0.946), *I. longirostris* (0.375–0.871) and *P. magdalenae* (0.399–0.949)
[23, 26, 28].

Results obtained do not support the hypothesis of low genetic variability levels in *C. magda-
lenae* as it showed high alleles/*locus* and $H_E$ mean values in relation to those reported for neo-
tropical Characiformes (Na: 10.92, $H_E$: 0.675) [81]. As this is the first study performed for
*Cyphocharax* genus, high diversity was found in comparison to *C. mivartii* (Na: 10.493, $H_O$:
0.757, $H_E$: 0.801), a Curimatidae species with which it shares its habitat [23]. Likewise, *C. mag-
dalenae* displayed greater values over the two estimators with respect to *I. longirostris* (Na:
8.84–11. 05, $H_O$: 0.701–0.767, $H_E$: 0.771–0.798; [28]) and similar values to those reported for *P.
magdalenae* (Na: 22.455, $H_O$: 0.725, $H_E$: 0.898; [26]) in the Cauca River. Even more, *C.

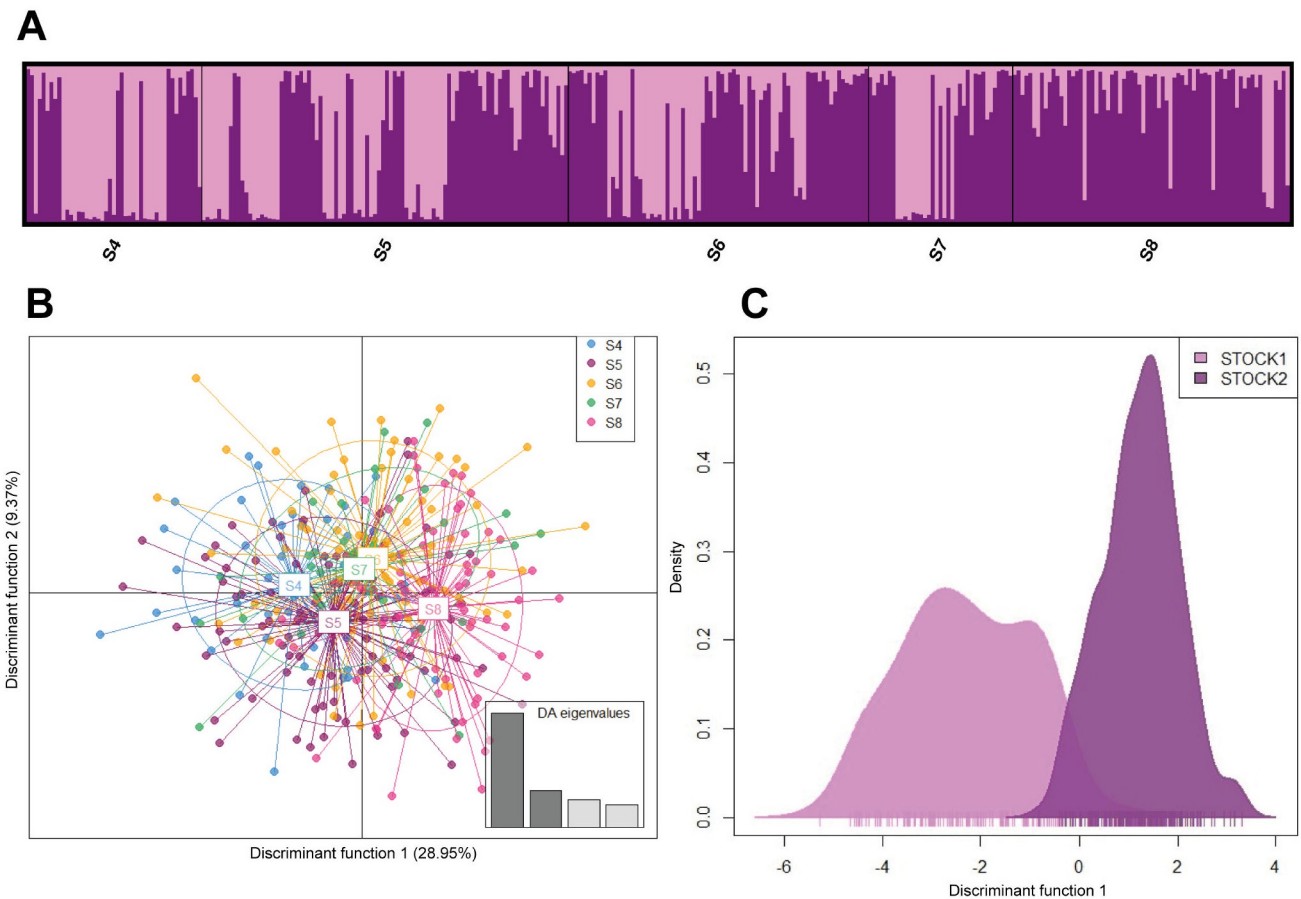

**Fig 3. Results of Bayesian inference of structure (a) and discriminant analysis of principal components of *Cyphocharax magdalenae*: (b) Five sections evaluated (c) Two genetic stocks.**

*magdalenae* showed greater values in alleles/*locus* and similar $H_E$ values compared to *P. magdalenae*, in rivers with great importance for Colombia, such as San Jorge (Na: 13.545, $H_O$: 0.809; $H_E$: 0.884), Sinú (Na: 15.273, $H_O$: 0.767, $H_E$: 0.882), Atrato (Na: 14.636, $H_O$: 0.718, $H_E$: 0.879), Nare (Na: 15.636, $H_O$: 0.659; $H_E$: 0.876), and Magdalena (Na: 19.455, $H_O$: 0.758; $H_E$: 0.896). *C. magdalenae* also exhibited much greater $H_E$ values than *P. reticulatus* ($H_E$: 0.400) even though this estimator was calculated based on a different molecular marker [27].

**Table 5. Pairwise comparisons of the genetic structure estimators of *Cyphocharax magdalenae*.** Jost's $D_{EST}$ (above diagonal) and $F'_{ST}$ (below diagonal).

| Pop | S4 | S5 | S6 | S7 | S8 |
|---|---|---|---|---|---|
| **S4** | - | 0.000 | 0.031* | 0.014 | 0.082* |
| **S5** | 0.005 | - | 0.019* | 0.007 | 0.042* |
| **S6** | 0.007* | 0.004* | - | 0.000 | 0.020* |
| **S7** | 0.008 | 0.006 | 0.006 | - | 0.037* |
| **S8** | 0.010* | 0.006* | 0.005* | 0.008* | - |

* Statistical significance.

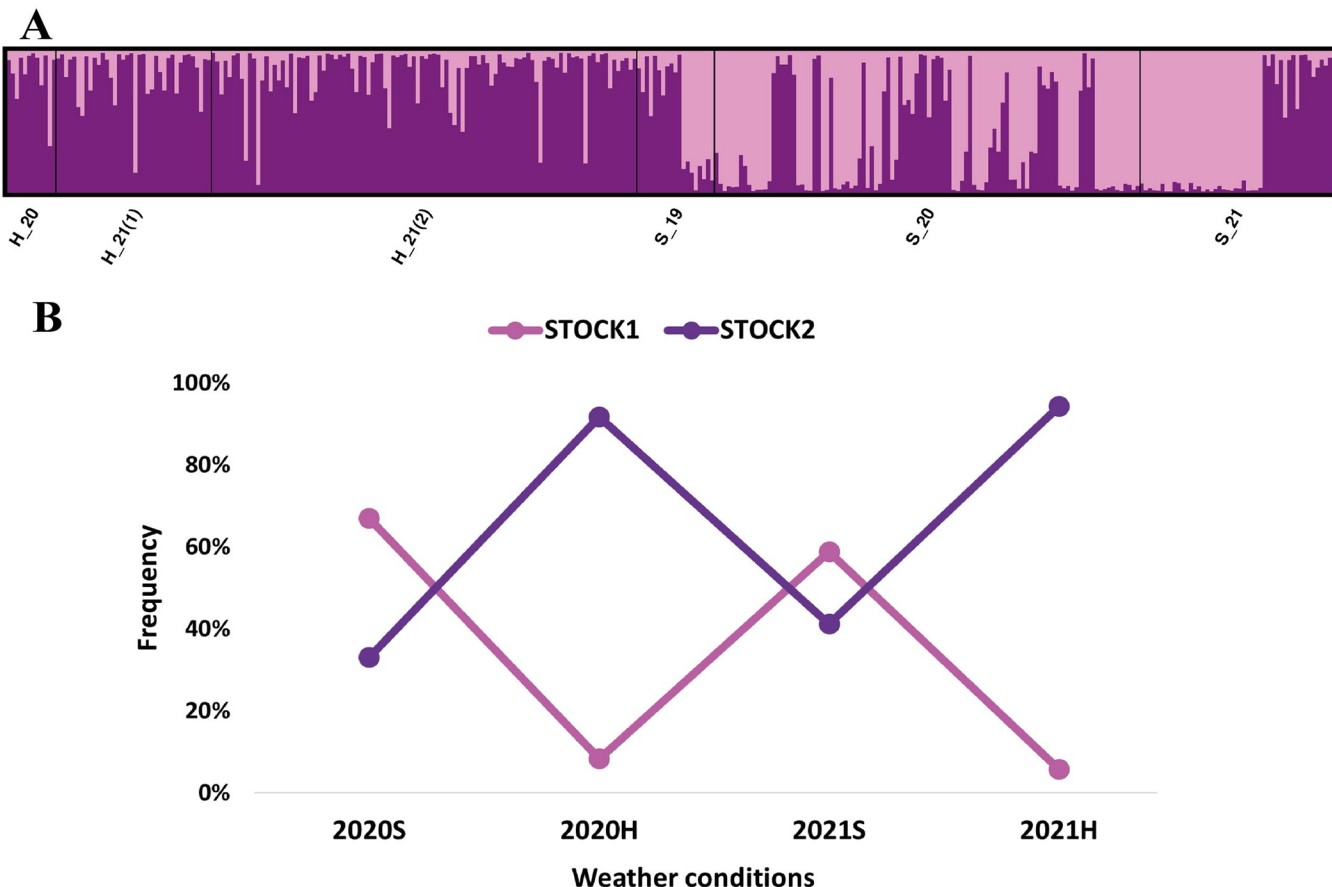

**Fig 4. Temporal structure of *Cyphocharax magdalenae* during 2019–2021 in the middle and lower sections of the Cauca River (A) and percentage of each stock by season and year of capture (B).** Rainy season (H): H_20: October 2020; H_21(1): April 2021; H_21(2): December 2021; Dry season (S): S_19: July-August-September 2019; S_20: February 2020; S_21: January 2021.

Furthermore, the hypothesis of high inbreeding levels in *C. magdalenae* was not confirmed since the species showed inbreeding signals below 10% in each genetic stock ($F_{IS}$ Stock1 = 0.051; $F_{IS}$ Stock2 = 0.071). Although these values do not exceed the proposed limit [82, 83], it has been indicated that any inbreeding coefficient greater than cero has unfavorable effects on the fitness [84]. Considering these results and those reported for Characiformes in the Magdalena-Cauca basin using species specific microsatellites, *C. magdalenae* exhibited inbreeding levels relatively similar to those reported for *C. mivartii* (0.040–0.087; [23]), lower values to those described for *P. magdalenae* (0.125–0.255; [26]), and greater to those found in *Brycon henni* (-0.040 – -0.009; [33]), a phylogenetically farther species.

Additionally, this study corroborated the hypothesis of high gene flow despite having found significant differences among individuals of the lower section of the Cauca River as those differences are related to the coexistence of two genetic stocks with uneven distribution through the river. This indicates that there is no evidence of barriers separating the gene flow of *C. magdalenae* in the lower section of the Cauca River, which is consistent with data reported for *C. mivartii* [23], *P. magdalenae* [26], *Cynopotamus magdalenae* (Steindachner 1879), *Megaleporinus muyscorum* [85] and catfishes *A. pardalis*, *P. grosskopfii*, *S. cuspicaudus* [86], *P. magdaleniatum* [30], *P. atricaudus* and *P. magnus* [36]. Nonetheless, genetic structure resulting from

seasonal variation was found since comparisons over the samples in rainy and dry seasons during the assessed years allowed detecting that one stock predominates in the dry season (Stock1) while the other predominates in the rainy season (Stock2). Sampling in both seasons for two years reflects an alternated cyclical behavior between both stocks and greater proliferative advantage of Stock2 over Stock1, which suggests a differential reproductive success leading to different cohorts (temporal Wahlund effect). These results are similar to those reported for *Prochilodus lineatus* (Valenciennes 1837) in the Mogi-Guaçu River in Brazil as individuals analyzed between 2005 and 2006 in rainy and dry seasons (January and August, respectively) showed evidence of temporal genetic structure [87]. Furthermore, it was found that *Salminus brasiliensis* (Cuvier 1816) in the Uruguay River in Brazil is formed by three genetic stocks resulting from a temporal genetic structure in which seasonal precipitations in the river are one of the main factors that could have originated genetic groups as the water level is fundamental in reproductive migration [88].

Furthermore, this study supports the hypothesis of a recent genetic bottleneck in *C. magdalenae*. These results may be a consequence of the joint action of various anthropogenic activities impacting the environment, directly affecting *C. magdalenae* individuals, or climatic changes. For instance, for The IUCN, *C. magdalenae* shows a decrease in its populations due to fishing pressure [11, 21, 22], which is consistent with the decline of this species reported in the Magdalena basin, with a diminishment in 887.7 tones unloaded for 2019 [21] in comparison to 436.1 tons for 2021 [22]. Moreover, there is no current regulation on a recommended or suggested minimum catch size for this species [89]; in fact, it was found that between 2018–2021 catch size ranges were lower than those described for size at maturity [22].

As it has been proposed for other species [86], contamination with mining-derived products could be affecting the viability of *C. magdalenae*, which is supported by evidence of methylmercury in fishes collected in the zone [45]. Another threat is the establishment of introduced exotic species [11, 90], such as Basa fish *Pangasianodon hypophthalmus*, a omnivorous species of fast size and weight growth, and having extension in Colombia, mainly in the Magdalena-Cauca basin [90], and the Nile Tilapia *Oreochromis niloticus*, which has the capacity to survive in all types of habitats, from salty, marine waters to estuarine and continental waters with temperatures between 8 and 42˚C [90]. Despite the introduction of *O. niloticus* in Colombia was done with commercial purposes and control of *Oreochromis mossambicus*, an aggressive behavior against native species *P. magdalenae* has been displayed [91] and, after its introduction, there has been a reduction in the abundance of *Triportheus magdalenae* (Steindachner 1878) in the Guajaro reservoir in Magdalena [92].

In addition to the anthropogenic activities, it has been documented that the impact of climate factors has provoked changes in fish populations as they are responsible for abrupt changes in environments such as high temperatures, alterations of hydrological cycles, dissolved oxygen reduction, changes in mortality rates, growth, reproduction and distribution of fish populations [93]. This climate factor has also been used for explaining population genetics aspects of *C. mivartii* [23] and *P. magdalenae* [26].

Contrary to the bottleneck evidence, estimations performed over $N_e$ (Stock1: 1464.6, Stock2: 2127) were higher than the critical value ($N_e \leq 1000$), which allows deducting that the species has long term maintenance [84]. In conjunction with the previously mentioned high genetic diversity levels, this result suggests that *C. magdalenae* populations are large enough to sustain elevated genetic diversity and $N_e$, despite exhibiting signals of bottleneck. This idea is also supported by the results of the conservation status test as even in 100 generations (260 years: species with generational time of 2.6 years) [19] estimations did not show evidence of heterozygosity reductions greater than 25% (critical value of the classification), for which the species can be listed as non-threatened. This classification, however, should be interpreted

carefully since it has been indicated that between 4–10% of freshwater species in South America are exposed to some extinction risk, mainly due to habitat loss or degradation [94].

To conclude, this study allowed determining that *C. magdalenae* in the lower section of the Cauca River exhibits seasonal population structure formed by two genetic stocks associated to the rainy and dry seasons, and which show high genetic diversity, low inbreeding coefficients, bottleneck signals and large $N_e$. Data obtained along with the microsatellite *loci* developed *de novo* in this study are a starting point for future research directed to monitoring the genetic diversity and population structure of this species to develop proper management plans.

## Supporting information

**S1 Table. Genotype data at 14 microsatellite *loci* included in the population genetic analysis of *Cyphocharax magdalenae*.** First row indicates respectively: Number of *loci*, Number of individuals, Number of sampling sections, Sample size for five sections. Third row indicates respectively: Sample ID, Sampling section, Genetic stock, Locus name for 14 *loci*.
(XLSX)

**S2 Table. Microsatellite loci not selected due to pitfalls in amplification or low levels of polymorphisms.** Na number of alleles per locus; $H_O$ Observed heterozygosity; $H_E$ Expected heterozygosity.
(XLSX)

## Acknowledgments

The authors express their gratitude to Universidad de Antioquia, Universidad de Córdoba, and Universidad Nacional de Colombia Sede Medellín for providing the fish tissue samples, and to the Proyecto CaVFish Colombia—Catálogo Visual de Peces de Agua Dulce de Colombia (cavfish.unibague.edu.co)—for providing the photograph of Fig 1.

## Author Contributions

**Conceptualization:** Ana Maria Ochoa-Aristizábal, Edna Judith Márquez.

**Data curation:** Ana Maria Ochoa-Aristizábal, Edna Judith Márquez.

**Formal analysis:** Ana Maria Ochoa-Aristizábal, Edna Judith Márquez.

**Funding acquisition:** Edna Judith Márquez.

**Investigation:** Ana Maria Ochoa-Aristizábal, Edna Judith Márquez.

**Methodology:** Ana Maria Ochoa-Aristizábal, Edna Judith Márquez.

**Project administration:** Edna Judith Márquez.

**Resources:** Edna Judith Márquez.

**Supervision:** Edna Judith Márquez.

**Validation:** Edna Judith Márquez.

**Visualization:** Ana Maria Ochoa-Aristizábal, Edna Judith Márquez.

**Writing – original draft:** Ana Maria Ochoa-Aristizábal, Edna Judith Márquez.

**Writing – review & editing:** Ana Maria Ochoa-Aristizábal, Edna Judith Márquez.

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
