## [Decision Letter · Decision Letter 0]

26 Dec 2023

PONE-D-23-37212Genetic insights into Cyphocharax magdalenae (Characiformes: Curimatidae): microsatellite loci development and population analysis in the Cauca River, ColombiaPLOS ONE

Dear Dr. Marquez,

Thank you for submitting your manuscript to PLOS ONE. After careful consideration, we feel that it has merit but does not fully meet PLOS ONE’s publication criteria as it currently stands. Therefore, we invite you to submit a revised version of the manuscript that addresses the points raised during the review process.

**ACADEMIC EDITOR: **

Please consider the following comments while revising your submission:

Some sentences are too long, which make the communication unclear.The manuscript needs to be proofread for English language and grammatical fixes by a proficient speaker.Other suggestions are indicated in the attached manuscript file.Please submit your revised manuscript by Feb 09 2024 11:59PM. If you will need more time than this to complete your revisions, please reply to this message or contact the journal office at plosone@plos.org. Please include the following items when submitting your revised manuscript:A rebuttal letter that responds to each point raised by the academic editor and reviewer(s). You should upload this letter as a separate file labeled 'Response to Reviewers'.A marked-up copy of your manuscript that highlights changes made to the original version. You should upload this as a separate file labeled 'Revised Manuscript with Track Changes'.An unmarked version of your revised paper without tracked changes. You should upload this as a separate file labeled 'Manuscript'.

We look forward to receiving your revised manuscript.

Kind regards,

Timothy Omara, PhD

Academic Editor

PLOS ONE

[This study was supported by a grant framed under the Project “Variabilidad genética de un banco de peces de los sectores medio y bajo del río Cauca” (CT-2019-000661, Empresas Públicas de Medellín and Universidad Nacional de Colombia, Sede Medellín).]

 [This study was supported by a grant framed under the Project “Variabilidad genética de un banco de peces de los sectores medio y bajo del río Cauca” (CT-2019-000661, Empresas Públicas de Medellín and Universidad Nacional de Colombia, Sede Medellín). Funders do not play any role in the study design, data collection and analysis, decision to publish, or preparation of the manuscript]

Additional Editor Comments:

Reviewers' comments:

Reviewer's Responses to Questions

**Comments to the Author**

1. Is the manuscript technically sound, and do the data support the conclusions?

Reviewer #1: Yes

Reviewer #2: Yes

2. Has the statistical analysis been performed appropriately and rigorously? 

Reviewer #1: Yes

Reviewer #2: Yes

3. Have the authors made all data underlying the findings in their manuscript fully available?

Reviewer #1: Yes

Reviewer #2: Yes

4. Is the manuscript presented in an intelligible fashion and written in standard English?

Reviewer #1: Yes

Reviewer #2: Yes

5. Review Comments to the Author

Reviewer #1: This paper aimed to develop the microsatellite marker and assessed the population structure of Cyphocharax magdalenae in Columbia. Authors did a great job and clearly interpret the data. However, this manuscript is too long and need to restructure as I suggested in the attached.

I did not find the results of AMOVA among the population of S4-S8. Did you calculate this AMOVA?

I also found something weird. You mention high genetic diversity and a recent genetic bottleneck. Bottleneck normally reduced genetic diversity. Why? I did not see you explain about this.

Cheers

Reviewer #2: This paper describes a routine process of developing microsatellite marker using NGS in Cyphocharax magdalenae, which has potential to become a baseline data for further studies.

Author may explain, why novel set of markers was developed in this study? Now a days there are plenty of the records of cross-species amplification. Was crass-amplification using other closely related species' microsatellite maker failed?

Though there were some discussion for remaining 16 markers. However, a supplementary file may be added explaining the statistics of remaining 16 loci.

A reference image of study species may be provided in paper.

Overall this is a crisp and clear methodology paper with basic population genetics of Cyphocharax magdalenae.

6. PLOS authors have the option to publish the peer review history of their article (what does this mean?). If published, this will include your full peer review and any attached files.

Reviewer #1: No

Reviewer #2: **Yes: **SANDEEP KUMAR GUPTA

---

## [Author Response · Author response to Decision Letter 0]

8 Feb 2024

Medellín, February 9th, 2024

Dr. Emily Chenette

Editor-in-chief PLOS ONE

Dear Editor 

We really appreciate the detailed revision of the reviewers and have edited the manuscript following your valuable recommendations, which have led to improve our paper.

We hope that the manuscript is now suitable for publication in Plos One.

Sincerely,

Edna J. Márquez

PONE-D-23-37212

Genetic insights into Cyphocharax magdalenae (Characiformes: Curimatidae): microsatellite loci development and population analysis in the Cauca River, Colombia

PLOS ONE

Dear Dr. Marquez,

Thank you for submitting your manuscript to PLOS ONE. After careful consideration, we feel that it has merit but does not fully meet PLOS ONE’s publication criteria as it currently stands. Therefore, we invite you to submit a revised version of the manuscript that addresses the points raised during the review process.

ACADEMIC EDITOR: 

Please consider the following comments while revising your submission:

Some sentences are too long, which make the communication unclear.

Done

The manuscript needs to be proofread for English language and grammatical fixes by a proficient speaker.

Done.

Other suggestions are indicated in the attached manuscript file.

Done

Done

Done

Please see below.

Complete information is documented in the manuscript.

We look forward to receiving your revised manuscript.

Kind regards,

Timothy Omara, PhD

Academic Editor

PLOS ONE

Done

[This study was supported by a grant framed under the Project “Variabilidad genética de un banco de peces de los sectores medio y bajo del río Cauca” (CT-2019-000661, Empresas Públicas de Medellín and Universidad Nacional de Colombia, Sede Medellín).]

 [This study was supported by a grant framed under the Project “Variabilidad genética de un banco de peces de los sectores medio y bajo del río Cauca” (CT-2019-000661, Empresas Públicas de Medellín and Universidad Nacional de Colombia, Sede Medellín). Funders do not play any role in the study design, data collection and analysis, decision to publish, or preparation of the manuscript]

Done. We have removed the text, and we would like to retain the current Funding Statement.

Now, we included the Supporting information “S1 Table”: “S1 Table. Genotype data at 14 microsatellite loci included in the population genetic analysis of Cyphocharax magdalenae. First row indicates respectively: Number of loci, Number of individuals, Number of sampling sections, Sample size for five sections. Third row indicates respectively: Sample ID, Sampling section, Genetic stock, Locus name for 14 loci.”

We include a new figure: “Fig 2. Sampling sites (stars) of Cyphocharax magdalenae in the lower section of the Cauca River (S4-S8). Self-made creation of the map based on contour lines scaled 1:100,000 from the Instituto Geográfico Agustín Codazzi source, 2019 (Available from: IGAC Geoportal, https://geoportal.igac.gov.co/contenido/datos-abiertos-cartografia-y-geografia).”

Additional Editor Comments:

Reviewers' comments:

Reviewer's Responses to Questions

Comments to the Author

1. Is the manuscript technically sound, and do the data support the conclusions?

Reviewer #1: Yes

Reviewer #2: Yes

2. Has the statistical analysis been performed appropriately and rigorously?

Reviewer #1: Yes

Reviewer #2: Yes

3. Have the authors made all data underlying the findings in their manuscript fully available?

Reviewer #1: Yes

Reviewer #2: Yes

4. Is the manuscript presented in an intelligible fashion and written in standard English?

Reviewer #1: Yes

Reviewer #2: Yes

5. Review Comments to the Author

Reviewer #1: This paper aimed to develop the microsatellite marker and assessed the population structure of Cyphocharax magdalenae in Columbia. Authors did a great job and clearly interpret the data. However, this manuscript is too long and need to restructure as I suggested in the attached.

Done

I did not find the results of AMOVA among the population of S4-S8. Did you calculate this AMOVA?

An AMOVA using all samples from each sector was performed to assess the genetic differentiation among all sections (see Summary AMOVA Table). The pair-wise comparisons, using the statistic F´ST and Jost´Dst are shown in the Table 5.

Summary AMOVA Table

Source df SS MS Est. Var. %

Among Sections 4 37.244 9.311 0.020 0%

Among Indiv 319 2148.416 6.735 0.537 9%

Within Indiv 324 1834.000 5.660 5.660 91%

Total 647 4019.660 6.218 100%

This information is summarized in the text: “…with results obtained in AMOVA (F´ST (4, 647) = 0.003, P = 0.001) and the pairwise comparisons of the standardized statistics F’ST, DEST (Table 5)”

I also found something weird. You mention high genetic diversity and a recent genetic bottleneck. Bottleneck normally reduced genetic diversity. Why? I did not see you explain about this.

Now, in the third line of the penultimate paragraph: “In conjunction with the previously mentioned high genetic diversity levels, this result suggests that C. magdalenae populations are large enough to sustain elevated genetic diversity and Ne, despite exhibiting signals of bottleneck.”

Cheers

Reviewer #2: This paper describes a routine process of developing microsatellite marker using NGS in Cyphocharax magdalenae, which has potential to become a baseline data for further studies.

Author may explain, why novel set of markers was developed in this study? Now a days there are plenty of the records of cross-species amplification. Was crass-amplification using other closely related species' microsatellite maker failed?

Now: “To test these hypotheses, this study identified de novo and characterized species-specific microsatellite loci to C. magdalenae. This approach was chosen instead of using heterologous loci to avoid potential pitfalls associated with cross-amplification [38], such as allele size homoplasy [39], unsuccessful amplification in phylogenetically distant species [40], low polymorphism, presence of null alleles [41], and amplification of non-orthologous loci [42]”

Though there were some discussion for remaining 16 markers. However, a supplementary file may be added explaining the statistics of remaining 16 loci.

Done. Now, we included the Supporting information “S2 Table”. “S2 Table. Microsatellite loci not selected due to pitfalls in amplification or low levels of polymorphisms. Na number of alleles per locus; HO Observed heterozygosity; HE Expected heterozygosity.”

A reference image of study species may be provided in paper.

Done. Now, we included a photography and the respective copyright permissions: “Fig 1. Photography of Cyphocharax magdalenae, a neotropical freshwater fish. Courtesy of Jorge E. García-Melo and Luis J. García-Melo, Proyecto CaVFish Colombia.”

Overall this is a crisp and clear methodology paper with basic population genetics of Cyphocharax magdalenae.

6. PLOS authors have the option to publish the peer review history of their article (what does this mean?). If published, this will include your full peer review and any attached files.

Do you want your identity to be public for this peer review? For information about this choice, including consent withdrawal, please see our Privacy Policy.

Reviewer #1: No

Reviewer #2: Yes: SANDEEP KUMAR GUPTA

---

## [Decision Letter · Decision Letter 1]

1 Apr 2024

Genetic insights into Cyphocharax magdalenae (Characiformes: Curimatidae): microsatellite loci development and population analysis in the Cauca River, Colombia

PONE-D-23-37212R1

Dear Dr. Marquez,

We’re pleased to inform you that your manuscript has been judged scientifically suitable for publication and will be formally accepted for publication once it meets all outstanding technical requirements.

Kind regards,

Timothy Omara, PhD

Academic Editor

PLOS ONE

Additional Editor Comments (optional):

Reviewers' comments:

Reviewer's Responses to Questions

**Comments to the Author**

1. If the authors have adequately addressed your comments raised in a previous round of review and you feel that this manuscript is now acceptable for publication, you may indicate that here to bypass the “Comments to the Author” section, enter your conflict of interest statement in the “Confidential to Editor” section, and submit your "Accept" recommendation.

Reviewer #1: All comments have been addressed

2. Is the manuscript technically sound, and do the data support the conclusions?

Reviewer #1: Yes

3. Has the statistical analysis been performed appropriately and rigorously? 

Reviewer #1: Yes

4. Have the authors made all data underlying the findings in their manuscript fully available?

Reviewer #1: Yes

5. Is the manuscript presented in an intelligible fashion and written in standard English?

Reviewer #1: Yes

6. Review Comments to the Author

Reviewer #1: (No Response)

7. PLOS authors have the option to publish the peer review history of their article (what does this mean?). If published, this will include your full peer review and any attached files.

Reviewer #1: No

---

## [Editor Report · Acceptance letter]

5 Apr 2024

PONE-D-23-37212R1 

PLOS ONE

Dear Dr. Márquez, 

I'm pleased to inform you that your manuscript has been deemed suitable for publication in PLOS ONE. Congratulations! Your manuscript is now being handed over to our production team.

Kind regards, 

on behalf of

Dr. Timothy Omara 

Academic Editor

PLOS ONE